# PAGE-B and REACH-B Predicts the Risk of Developing Hepatocellular Carcinoma in Chronic Hepatitis B Patients from Northeast, Brazil

**DOI:** 10.3390/v14040732

**Published:** 2022-03-31

**Authors:** Alessandra Porto de Macedo Costa, Marcos Antonio Custódio Neto da Silva, Rogério Soares Castro, Ana Leatrice de Oliveira Sampaio, Antônio Machado Alencar Júnior, Márcia Costa da Silva, Adalgisa de Souza Paiva Ferreira

**Affiliations:** 1Center for Liver Studies, University Hospital of the Federal University of Maranhão, Saint Louis CEP 65020-070, MA, Brazil; alebisio@uol.com.br (A.P.d.M.C.); rogeriocastro79@hotmail.com (R.S.C.); leatrice_samp@hotmail.com (A.L.d.O.S.); aljr1810@gmail.com (A.M.A.J.); 2Federal University of Maranhão, Medicine Course, Saint Louis CEP 65020-070, MA, Brazil; 3Epidemiological Surveillance Service, State Health Department, Saint Louis CEP 65076-820, MA, Brazil; marcia.silva@ufma.br

**Keywords:** chronic hepatitis B, hepatocellular carcinoma, prevalence, surveillance, PAGE-B, REACH-B

## Abstract

This study aims to evaluate the accuracy of the PAGE-B and REACH-B scores in predicting the risk of developing HCC in patients with chronic hepatitis B regularly followed up at a reference service in the State of Maranhão. A historical, longitudinal, retrospective cohort study, carried out from the review of medical records of patients with chronic Hepatitis B. PAGE-B and REACH-B scores were calculated and the accuracy of the scores in predicting the risk of HCC in the studied population was evaluated. A total of 978 patients were included, with a median age of around 47 years, most of them female and not cirrhotic. HCC was identified in 34 patients. Thrombocytopenia, high viral load, male gender and age were associated with the occurrence of HCC. The ROC curve for the PAGE-B score showed a value of 0.78 and for the REACH-B score of 0.79. The cutoff point for PAGE-B was 11 points for greater sensitivity and for REACH-B 7.5 points considering greater sensitivity and 9.5 points considering greater specificity. PAGE-B and REACH-B scores were able to predict the risk of developing HCC in the studied population. The use of risk stratification scores is useful to reduce costs associated with HCC screening.

## 1. Introduction

Chronic hepatitis B virus (HBV) infection is the leading cause of cirrhosis, liver failure and hepatocellular carcinoma (HCC) worldwide [1]. Of chronic carriers of HBV, approximately 15–40% develop chronic hepatitis B [2]. About 90% of patients with chronic hepatitis B seroconvert from HBeAg to anti-HBe and become inactive carriers. However, approximately 10% of patients with chronic hepatitis B have active hepatitis and develop liver cirrhosis at a rate of 2% per year. As the progression of liver disease in patients with chronic hepatitis B is closely associated with active viral replication, a high level of HBV DNA is known to be an independent risk factor for disease progression. Therefore, suppression of HBV with antiviral therapy may reduce the risk of developing cirrhosis and HCC [3].

The risk factors for disease progression in patients with chronic hepatitis B can be divided into three categories: host-related factors, viral factors and hepatic factors [4,5,6]. Host-related factors can be modifiable or not, such as male gender, age, obesity, family history of HCC, cirrhosis, diabetes mellitus, smoking, alcohol, genetic predisposition, among others. Viral factors include high levels of viral load (HBV-DNA), HBV genotype, high serum levels of surface antigen (quantitative HBsAg), and presence of hepatitis B virus “E” antigen (HBeAg) [7,8,9,10,11,12,13]. Liver factors consist especially of the presence of liver cirrhosis, but also the presence of active hepatitis and coinfection with the hepatitis C virus and the presence of alcoholic or non-alcoholic fatty liver disease [6,14,15,16,17].

To assess the risk of patients with chronic hepatitis B to develop HCC, several prognostic scores were developed based on clinical and laboratory parameters, which can help the physician to improve the efficiency and implementation of surveillance and screening strategies [7]. There are scores that were developed in untreated patients, such as the following: GAG-HCC (guide with age, sex, HBV DNA, central promoter mutations and cirrhosis-hepatocellular carcinoma), NGM-HCC (nomogram-hepatocellular carcinoma) and REACH-B(risk estimate for hepatocellular carcinoma in chronic hepatitis B); recent models constructed in treated patients include the following: modified REACH-B (mREACH-B), PAGE-B (platelets, age, sex, and HBV), modified PAGE-B (mPAGE-B), CAMD (cirrhosis, age, male sex, and diabetes mellitus) and REAL-B (Asia-Pacific Rim real-world efficacy for HBV risk scoring); other models based on mixed patients with a proportion of different treatment include the following: CU-HCC (Chinese University-hepatocellular carcinoma), LSM-HCC (liver stiffness measurement-hepatocellular carcinoma) and RWS-HCC (real-world risk score-hepatocellular carcinoma). However, none of them were recommended by guidelines to be widely used in clinical practice. Some crucial case combinations between these models, such as cirrhosis ratio and baseline alanine aminotransferase (ALT) level, appeared significantly different [6,14,18,19,20,21,22,23,24,25,26,27].

Most of the scores currently used have been validated in Asian patients. There are scores that are used for patients undergoing treatment for chronic HBV infection and others that are validated for untreated patients. Studies including European Caucasian and American patients have shown that the models are slightly less predictive; however, rates of HCC were very low, significantly limiting conclusions [28,29].

Given this context, the objective of this study was to evaluate the accuracy of the PAGE-B and REACH-B scores in predicting the risk of developing HCC in patients with chronic hepatitis B regularly followed up at a reference service in the State of Maranhão, with a view to improving the efficiency of HCC screening and surveillance in this population.

## 2. Materials and Methods

### 2.1. Study Type

This is a historical, longitudinal, retrospective cohort study, carried out from the review of medical records of patients with chronic Hepatitis B followed up at the Liver Studies Center of the University Hospital of the Federal University of Maranhão.

### 2.2. Study Population

The population selected for this study consisted of all patients with a confirmed diagnosis of chronic Hepatitis B who are being regularly monitored at the Liver Center of the University Hospital of the Federal University of Maranhão since the beginning of activities.

Inclusion criteria were: patients of both sexes, with positive HBsAg and total anti-HBc for at least six months and with complete clinical and laboratory data and with at least 3 follow-up visits with a maximum interval of 1 year between consultations.

Exclusion criteria were: patients without complete information in the medical records or with less than 3 follow-up visits with a maximum interval of 1 year or 3 follow-up visits in the same year.

### 2.3. Sample

The sample consisted of 1398 patients with chronic hepatitis B of both sexes. A total of 420 patients were excluded, according to the exclusion criteria, with a final sample of 978 patients with chronic hepatitis B.

To calculate the ROC curve of the PAGE-B and REACH-B scores, patients with at least 5 and 3 years of follow-up, respectively, were considered.

### 2.4. Definitions

#### 2.4.1. Stages of Chronic HBV Infection

The stages of chronic HBV infection were defined according to the recommendations of the European Association for the Study of the Liver (EASL, 2017) [30].

Chronic HBeAg positive HBV infection (immunotolerant): presence of HBeAg, high levels of HBVDNA (>107 IU/mL) and ALT persistently within normal reference values (approximately 40 IU/mL). Histology with minimal or no necroinflammatory activity or fibrosis;Chronic hepatitis B HBeAg positive (immunoactive): presence of HBeAg, elevated HBVDNA (104–107 IU/mL) and ALT above the normal reference value. Histology with moderate or intense necroinflammatory activity and fibrosis;Chronic HBeAg negative HBV infection (inactive carrier): presence of anti-HBe antibody, undetectable or low HBVDNA (<2000 IU/mL), and normal ALT. Histology with minimal necroinflammatory activity;Chronic hepatitis B HBeAg negative (reactivation): absence of HBeAg, usually with the presence of antiHBe, high levels of HBVDNA (>2000 IU/mL) and ALT above the normal reference value (persistently or intermittently). Histology with moderate or intense necroinflammatory activity and fibrosis.

#### 2.4.2. Liver Cirrhosis

The presence of cirrhosis was defined by clinical, histological, radiological (liver elastography, ultrasonography, computed tomography, magnetic resonance), endoscopic and laboratory criteria.

According to the presentation of cirrhosis, patients were classified into:

Compensated: absence of symptoms of decompensated cirrhosis.

Decompensated: presence of current or past signs and/or symptoms of decompensated cirrhosis: jaundice, ascites, gastrointestinal bleeding and encephalopathy.

#### 2.4.3. Diagnosis of Hepatocellular Carcinoma

The diagnosis of HCC was identified through the results of imaging tests, alpha-fetoprotein dosage and histopathology recorded in the medical record.

#### 2.4.4. Criteria for the Treatment of Chronic Hepatitis B

Patients started to receive entecavir (ETV) or tenofovir (TDF) based on serum HBV-DNA levels and severity of liver disease according to the criteria of the Clinical Protocol and Therapeutic Guidelines for Chronic Hepatitis B of the Brazilian Ministry of Health. The criteria for starting therapy were as follows: patient with reactive HBeAg and ALT > 2× upper limit of normal (ULN); adult over 30 years of age with HBeAg reagent; Patient with non-reactive HBeAg, HBV-DNA >2000 IU/mL and ALT > 2× ULN [31].

Other criteria independent of HBeAg included: family history of HCC; extrahepatic manifestations with disabling motor involvement, arthritis, vasculitis, glomerulonephritis and polyarteritis nodosa; HIV/HBV or HCV/HBV coinfection; severe acute hepatitis (coagulopathies or jaundice for more than 14 days); chronic hepatitis B reactivation; cirrhosis/liver failure; liver biopsy METAVIR ≥ A2F2 or liver elastography > 7.0 kPa; prevention of viral reactivation in patients who will receive immunosuppressive therapy (IMSS) or chemotherapy (CT) [31].

### 2.5. Data Collection

Data were obtained from the analysis of patients’ charts.

The following data were analyzed: age, gender, race, origin, alcohol consumption, comorbidities, co-infection with hepatitis C virus and/or HIV, positive family history for HCC, pre- and post-treatment HBVDNA levels, serological profile for HCV antigen. replication, ALT levels, alpha-fetoprotein, platelets, presence of cirrhosis and abdominal ultrasound.

In addition, treatment time, antivirals used, dates of consultations and HCC diagnosis were evaluated.

From these data, the PAGE-B and REACH-B scores were calculated to predict the risk of hepatocellular carcinoma in the studied population and to determine the accuracy of these scores in this population.

### 2.6. Follow-Up

Patients underwent abdominal US ± serum AFP measurement every 6–12 months for HCC surveillance. When any new lesions were detected on US, patients underwent triphasic CT or dynamic MRI. The diagnosis of HCC was made according to guidelines.

Follow-up was the time interval between initiation of therapy and diagnosis of HCC or the last visit with an imaging result available in the absence of HCC.

Serum levels of HBV DNA were measured by PCR. HBV DNA negativity was considered when the serum level of HBV DNA was <80 IU/mL. Sustained virological response was considered once HBV-DNA negativity was achieved and maintained throughout the course of therapy.

### 2.7. PAGE-B and REACH-B Scores

The PAGE-B score is based on patient age, sex and platelets and represents a simple and reliable score for predicting 5-year HCC risk in Caucasian patients with chronic B virus infection on entecavir/tenofovir. The score ranges from 0 to 25 points. Low risk patients are those with ≤9 points, intermediate risk between 10–17 points and high risk ≥18 points [22].

The REACH-B score assesses the risk of developing HCC at 3, 5 and 10 years in untreated patients. The variables included in the risk score are sex, age, ALT, HBeAg status and HBVDNA. The maximum score is 17 points. This score was first validated in the Asian population. The interpretation of the score is described in Table 1 according to Yang et al. (2011) [6].

### 2.8. Statistical Analysis

Exploratory data analysis included descriptive statistics, mean, median, standard deviation, minimum and maximum values for numerical variables and number and proportion for categorical variables. To analyze the behavior of continuous variables, descriptive statistics, histogram and boxplot plots and the specific test for the theoretical assumption of normality Shapiro–Wilk were considered.

The comparison between two groups was performed using Student’s *t* test or Mann–Whitney test for numerical variables and Pearson’s chi-square test or Fisher’s exact test for categorical variables.

ROC curves for PAGE-B and REACH-B scores were constructed in predicting the HCC outcome; the cutoff point that maximized sensitivity and specificity was determined to calculate the accuracy of the marker: sensitivity (S), specificity (E), positive predictive value (PPV), negative predictive value (VPN), positive likelihood ratio (PVR), negative likelihood ratio (RVN).

Time-dependent ROC curve was also constructed for the variables PAGE-B and REACH-B, in the prediction of HCC over time, using the R software, survivalROC package.

The ROC curve (receiver operating characteristics) is a traditional and very common method to evaluate the performance of a model for a given event when the response variable is binary. However, when the studied event contains censored and time-dependent data, the most appropriate method is the ROC curve as a function of survival time.

In this perspective, Heagerty and Zheng (2005) [32] present a method of temporal variation of sensitivity and specificity, being considered an incident/dynamic according to the mathematical formulation:(I)*ensi*tivity (*c*, *t*): *Pr* {*Mi* > *c* | δ*i* (*t*) =} = *Pr* {*Mi* > *c* | *Ti* = *t*}(II)*speci*ficity (*c*, 𝑡): *Pr* {*Mi* ≤ *c* | δ*i* (*t*) =} = *Pr* {*Mi* ≤ *c* | *Ti* > *t*}

On what:M = continuous markerc = truncationt = instant of time*δ*(*t*) = II(*T* ≤ *C*): failure or censoring event indicator, considering C the censoring time and T the survival time.

Statistical analysis was performed using IBM-SPSS Statistics software version 28 (IBM Corporation, NY, USA) and R software (R CORE TEAM, 2015). *p* values < 0.05 were considered significant.

### 2.9. Ethical Aspects

This project was submitted and approved by the Research Ethics Committee of the University Hospital of the Federal University of Maranhão under number 23523.006485/2017-06.

## 3. Results

### 3.1. Epidemiological and Clinical Characteristics

We included 978 patients with chronic hepatitis B regularly followed up at the Liver Center of the University Hospital of UFMA.

Table 2 describes clinical and epidemiological characteristics of the patients included. The median age was 47 years, most were female (54.3%), had no history of alcohol consumption (57.4%), did not have diabetes (90.4%) or hypertension (80. 4%). Most patients with chronic hepatitis B were not cirrhotic (81.4%). As for the family history of HBV and HCC, only 17.8% and 1.7% of the patients, respectively, had a positive family history. (Table 2).

Table 3 shows the serological profile, viral load and laboratory data of the included patients. As for the replication profile, 131 (13.4%) of the patients were HBeAg positive and the majority (75.2%) were anti-HBe positive. Most patients had a viral load < 300 (41.2%) and platelets ≥ 200,000 (58.4%). The median of ALT/ULN values was 0.63.

Among chronic hepatitis B patients, 386 (39.5%) met the criteria for treatment with antivirals. Of these, most were using tenofovir (48.9%) and entecavir (48.4%). (Table 4).

### 3.2. Prevalence of HCC and Univariate Analysis of Risk Factors

The prevalence of HCC in the population studied was 34 cases (3.5%), and most of the diagnosis was made through imaging tests, such as magnetic resonance imaging alone (53.5%) or in combination with alpha-fetoprotein dosage (20.6%), after screening ultrasound suggests the presence of a suspicious nodule.

Table 5 describes the clinical and epidemiological factors associated with the diagnosis of HCC in the population studied. Male sex, older age, presence of cirrhosis, presence of diabetes and not having a family history of HBV were associated with the diagnosis of HCC.

Table 6 describes the association between serological variables, viral load and laboratory tests with the occurrence of HCC. HBeAg positive, anti-HBe negative, thrombocytopenia and high viral load were associated with the occurrence of HCC.

### 3.3. Risk Scores for HCC

To assess the risk of developing HCC in the studied population, two prognostic scores were applied, PAGE-B and REACH-B. To analyze the accuracy of the PAGE-B, only patients with medical follow-up for at least 5 years (*n* = 479) were included. To analyze the accuracy of REACH-B, only patients with medical follow-up for at least 3 years (*n* = 682) were included.

The median score for PAGE-B was 10 and for REACH-B it was 6.

### 3.4. Application of the Time-Dependent ROC Curve Technique—ROC (t)

According to the mathematical formulation describe in the Methods section, for each instant t, the units at risk are initially divided into two groups. Sensitivity is measured with a group of individuals with a mark greater than the truncation (c) and the group of individuals referring to specificity, composed of individuals that have a time of failure or censorship greater than the instant of time (t). Based on this approach, it is possible to estimate the time dependent ROC curve. The graphic representation deals with ordered pairs: “sensitivity (c, t)” versus “1—specificity (c, t)”, for all values of c. From this approach, one can also calculate the area under the ROC curve, corresponding to AUC, where the value 1 represents the perfect model, while an area below 0.50 represents the model with poor data adherence. In addition, AUC measures the predictive ability of the model to correctly classify individuals in an event [33].

To generate the results of the time-dependent ROC curve, the R software package known as survivalROC was used, constructing the graphs of the PAGE-B and REACH-B variables that indicated the behavior of sensitivity and 1-specificity, over a given time. (t). The results are shown in Figure 1 and Figure 2 below and in Table 7.

It is observed that considering the scenario of times t = 1, 2 and 3 years, the values of AUC(t) in both graphs showed an increase and improvement in the shape of the curve, becoming more concave, expressing the idea of data adherence. Consequently, the results imply the most appropriate values of sensitivity and specificity.

### 3.5. PAGE-B and REACH-B ROC Curve

To analyze the performance of the PAGE-B and REACH-B scores in the prediction of HCC, initially the ROC curve was constructed with the values of sensitivity and complement of specificity.

To analyze the accuracy of the PAGE-B, only patients with medical follow-up for at least 5 years (*n* = 479) were included. Of the 479 patients, 8 developed HCC.

To analyze the accuracy of REACH-B, only patients with medical follow-up for at least 3 years (*n* = 682) were included. Of the 682 patients, 16 developed HCC.

The ROC curve is in the figures below (Figure 3 and Figure 4).

The cut-off point that maximized sensitivity and specificity for PAGE-B was 11 points. In this way, the accuracy was calculated, as shown in the table below (Table 8).

For the REACH-B score, two cut-off points were interesting to maximize sensitivity and specificity, with 7.5 showing greater sensitivity and 9.5 greater specificity. Thus, the accuracy was calculated for the two cut-off points (Table 9 and Table 10).

## 4. Discussion

Patients with chronic hepatitis B are at risk of developing HCC, with a significant impact on morbidity and mortality associated with this diagnosis [34]. In this sense, screening every 6–12 months with ultrasound associated with alpha-fetoprotein dosage is recommended in patients with chronic hepatitis B, especially in those with liver cirrhosis [35,36].

However, later studies identified patients with a cumulatively lower incidence of developing HCC, especially patients undergoing treatment with nuclei(t)id analogues [3,37]. This uncertainty of residual HCC development in treated patients was addressed by the PAGE-B risk score (integrating age, sex, and platelet count, which primarily selects Caucasian patients at low risk for HCC [22,38,39]. PAGE-B score, patients in the low-risk HCC group (≤9 points) do not develop HCC on stable therapy during a 5-year follow-up [22]. These data led the European Association for the Study of the Liver (EASL) to recommend that patients with chronic hepatitis B categorized as low risk by PAGE-B could delay the HCC surveillance strategy [40].

In our series, we identified a good sensitivity of the PAGE-B score in predicting the risk of developing HCC at 5 years of follow-up. For our population, we identified the value of 11 points as having the best sensitivity and accuracy for diagnosing HCC in the included patients. The analysis of the ROC curve (t) also identified better AUC as the years of follow-up increased. Thus, it is suggested that patients with less than 11 points on the PAGE-B score may have a longer follow-up time than traditionally performed.

A study carried out by Sprinzl et al. (2021) showed that the adoption of the PAGE-B score in patients with Hepatitis B as a screening tool could reduce screening costs for HCC by 15.51%. In the population studied in Germany, 1.35% to 7.65% of patients of Caucasian descent infected with HBV could defer HCC screening according to population estimates if they adopted PAGE-B for risk stratification [41].

Wu et al. (2021) performed a meta-analysis evaluating several prognostic scores on HCC. Discrimination was generally acceptable for all scores analyzed, with an area under the curve ranging from 0.70 (for REACH-B: 95% CI, 0.63–0.76) for a 3-year forecast, 0.68 (REACH-B: 95% CI, 0.64–0.73) for a 5-year forecast and 0.70 (PAGE-B: 95% CI, 0.58–0.80) to 0.81 for a 10-year forecast [42]. In the analyzed cohort, AUC for PAGE-B was 0.78 and REACH-B was 0.79, according to literature data.

Several HCC risk scores have been developed to predict the risk of HCC in patients with chronic hepatitis B. All of these risk scores use clinical variables and appear to be easily applicable for most patients. Direct comparison between different scores is not possible due to differences in the variables analyzed. Although different HCC risk scores show variable performance in different populations, they all have high NPVS to exclude the development of HCC in patients with chronic hepatitis B. To date, PAGE-B demonstrates good predictability for the development of HCC in patients, with Asians and Caucasians undergoing treatment [43].

Some limitations of the two scores can also be highlighted. A significant decrease in discrimination in REACH-B was detected for treated vs. untreated patients (p 1⁄4 0.004 for a 5-year forecast and *p* < 0.001 for a 10-year forecast) and cirrhotic vs. non-cirrhotic patients (p 1⁄4 0.003 for a 3-year forecast). Regarding calibration, underestimation of the risk of HCC was detected in REACH-B (3-year total O:E ratio 1⁄4 2.58) and PAGE-B (5-year total O:E ratio 1⁄4 1 0.70) for cirrhotic patients).

## 5. Conclusions

Patients with chronic hepatitis B are at risk for developing HCC. In our series, most patients were cirrhotic. The performance of PAGE-B and REACH-B scores proved to be an interesting tool to predict the risk of developing HCC in the studied population, being easy to apply and reproducible.

The PAGE-B and REACH-B scores showed good results in predicting the development of HCC in our cohort. The use of these scores may contribute to reducing the costs of performing ultrasound and laboratory tests to screen for HCC.

## Figures and Tables

**Figure 1 viruses-14-00732-f001:**
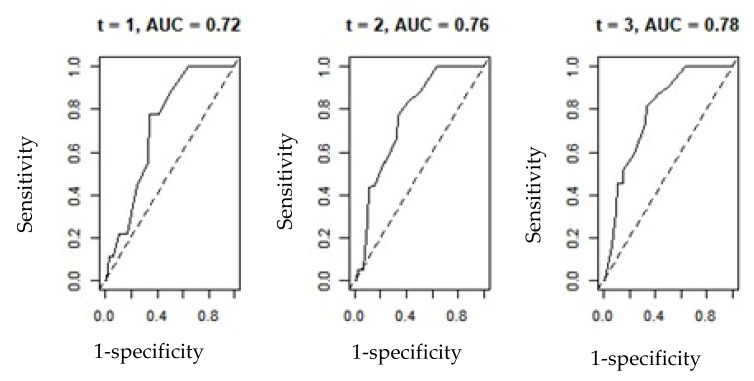
ROC curve-dependent time for the PAGE-B variable.

**Figure 2 viruses-14-00732-f002:**
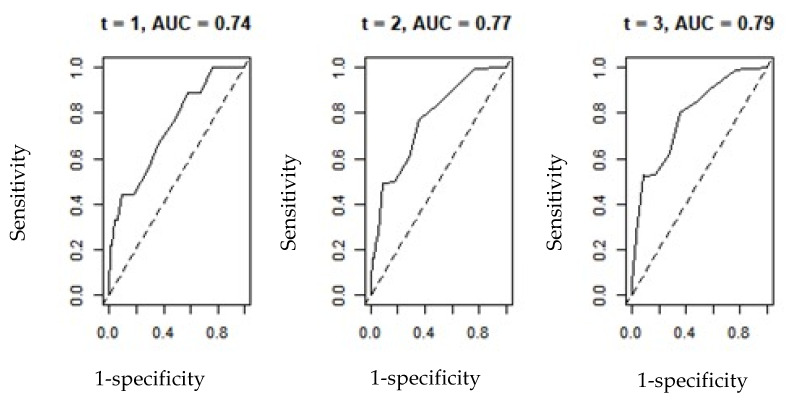
ROC curve-dependent time for the REACH-B variable.

**Figure 3 viruses-14-00732-f003:**
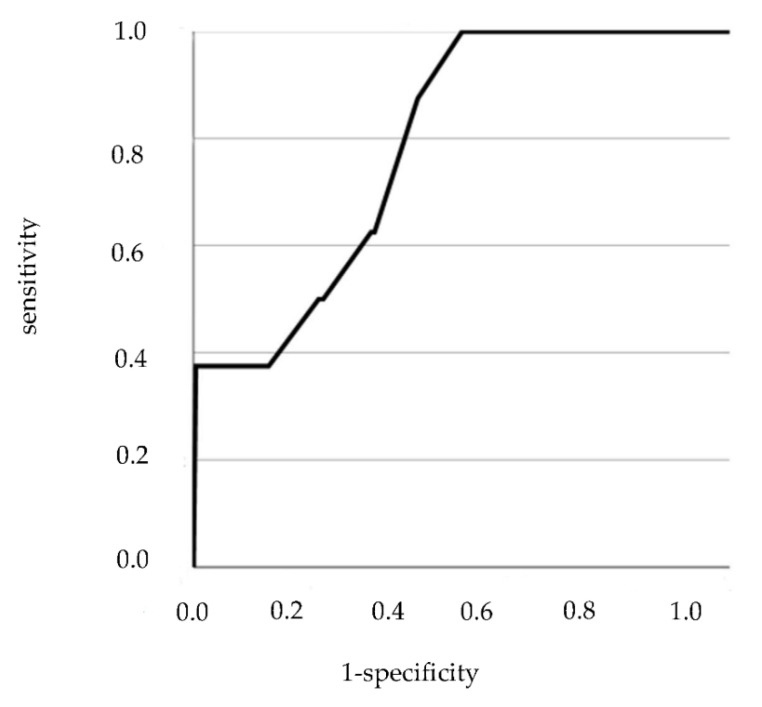
ROC curve for the PAGE-B variable.

**Figure 4 viruses-14-00732-f004:**
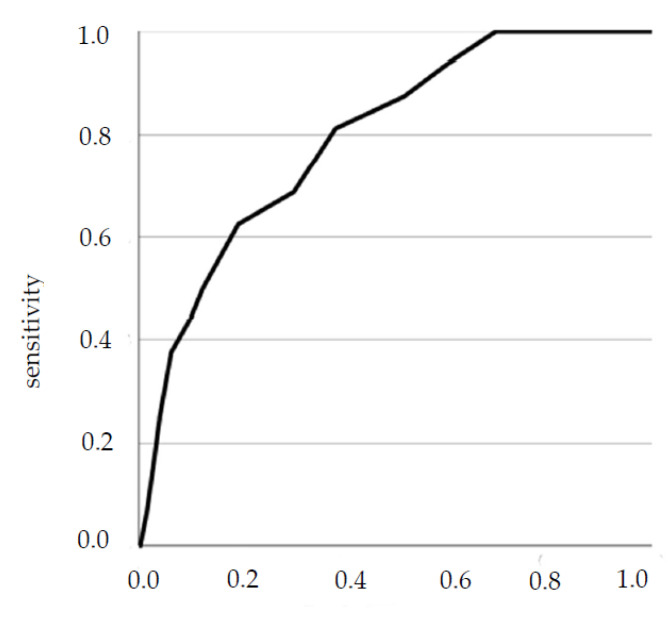
ROC curve for the REACH-B variable.

**Table 1 viruses-14-00732-t001:** REACH-B interpretation score (adapted).

Points	3 Years Risk	5 Years Risk	10 Years Risk
0	0.0%	0.0%	0.0%
1	0.0%	0.0%	0.1%
2	0.0%	0.0%	0.1%
3	0.0%	0.1%	0.2%
4	0.0%	0.1%	0.3%
5	0.1%	0.2%	0.5%
6	0.1%	0.3%	0.7%
7	0.2%	0.5%	1.2%
8	0.3%	0.8%	2.0%
9	0.5%	1.2%	3.2%
10	0.9%	2.0%	5.2%
11	1.4%	3.3%	8.4%
12	2.3%	5.3%	13.4%
13	3.7%	8.5%	21.0%
14	6.0%	13.6%	32.0%
15	9.6%	21.3%	46.8%
16	15.2%	32.4%	64.4%
17	23.6%	47.4%	81.6%

**Table 2 viruses-14-00732-t002:** Epidemiological and clinical characteristics of the 978 patients with chronic hepatitis B included.

Variables	*N* (%)
**Age** **(years)** *****	47.0 (14.0–92.0)
**Gender**	
Female	531 (54.3)
Male	447 (45.7)
**Alcohol**	
No	561 (57,4)
Yes	417 (42,6)
**Diabetes** **(DM)**	
No	884 (90.4)
Yes	94 (9.6)
**Hypertension** **(HAS)**	
No	786 (80.4)
Yes	192 (19.6)
**Cirrhosis**	
No	795 (81.3)
Yes	183 (18.7)
**Family** **history** **of** **HBV**	
No	804 (82.2)
Yes	174 (17.8)
**Family** **history** **of** **HCC**	
No	961 (98.3)
Yes	17 (1.7)

* Data described in median.

**Table 3 viruses-14-00732-t003:** Serological profile, viral load and laboratory tests among the 978 cases included in the study.

Variables	*N* (%)
**HBsAg**	
Positive	978 (100)
**Total** **Anti-HBc**	
Positive	978 (100)
**HBeAg**	
Negative	847 (86.6)
Positive	131 (13.4)
**Anti-HBe**	
Negative	243 (24.8)
Positive	735 (75.2)
**HBV-DNA**	
<300	403 (41.2)
300–9.999	351 (35.9)
10.000–99.999	73 (7.5)
100.000–999.999	37 (3.7)
>1000.000	114 (11.7)
**Platelets**	
≥200.000	571 (58.4)
100.000–199.999	329 (33.6)
<100.000	78 (8.0)
**ALT** **(xULN)** *****	0.63 (0.12–12.80)

* Data described in median.

**Table 4 viruses-14-00732-t004:** Data related to treatment of the 978 cases included.

Variables	*N* (%)
**Treatment**	
No	592 (60.5)
Yes	386 (39.5)
**Tenofovir**	189 (48.9)
**Entecavir**	187 (48.4)
**Adefovir**	01 (0.3)
**Lamivudine**	04 (1.1)
**Interferon**	05 (1.3)

**Table 5 viruses-14-00732-t005:** Univariate analysis of factors associated with the occurrence of HCC in the 978 patients included in the study.

	HCC		
Variables	No *n* (%)	Yes *n* (%)	*p*-Value
**Gender**			
Female	521 (98.1)	10 (1.9)	**0.003** *
Male	423 (94.6)	24 (5.4)	
**Age** **(years, median)**			
Median	47.0	60.5	**<0.001** ^@^
**Alcohol**			
No	541 (96.4)	20 (3.6)	0.861 *
Yes	403 (96.6)	14 (3.4)	
**Diabetes**			
No	857 (96.9)	27 (3.1)	**0.037** ^#^
Yes	87 (92.6)	07 (7.4)	
**Hypertension**			
No	762 (96.9)	24 (3.1)	0.144 *
Yes	182 (94.8)	10 (5.2)	
**Cirrhosis**			
No	795 (100.0)	00 (0.0)	**<0.001** *
Yes	149 (81.4)	34 (18.6)	
**Family** **history** **of** **HBV**			
No	770 (95.8)	34 (4.2)	**0.006** *
Yes	174 (100.0)	0 (0.0)	
**Family** **history** **of** **HCC**			
No	927 (91.8)	34 (3.5)	0.545 ^#^
Yes	17 (100.0)	0 (0.0)	

*: Pearson’s chi-square test; ^@^: Mann–Whitney test; ^#^: Fisher’s exact test.

**Table 6 viruses-14-00732-t006:** Univariate analysis of serological factors, viral load and laboratory tests associated with the occurrence of HCC in the 978 patients included in the study.

	HCC		
Variables	No *n* (%)	Yes *n* (%)	*p*-Value
**HbeAg**			
Negative	828 (97.8)	19 (2.2)	**<0.001** ^#^
Positive	116 (88.5)	15 (11.5)	
**anti-HBe**			
Negative	228 (93.8)	15 (6.2)	**0.008** *
Positive	716 (97.4)	19 (2.6)	
**Platelets**			
≥200.000	564 (98.8)	07 (1.2)	**<0.001** *
199.999–100.000	313 (95.1)	16 (4.9)	
<100.000	67 (85.9)	11 (14.1)	
**HBVDNA**			
≤300	389 (96.5)	14 (3.5)	**0.007** ^#^
300–9.999	345 (98.3)	06 (1.7)	
10.000–99.999	69 (94.5)	04 (5.5)	
100.000–999.999	32 (86.5)	05 (13.5)	
>1000.000	109 (95.6)	05 (4.4)	

*: Pearson’s chi-square test; ^#^: Fisher’s exact test.

**Table 7 viruses-14-00732-t007:** Area under the ROC curve of PAGE-B and REACH-B markers in HCC prediction.

		CI 95%		
Score	Area	Lower Limit	Upper Limit	*p*-Value
PAGE-B	0.788	0.661	0.915	**0.005**
REACH-B	0.794	0.695	0.893	**<0.001**

**Table 8 viruses-14-00732-t008:** Distribution of patients with and without hepatocellular carcinoma, according to the PAGE-B score and calculation of accuracy.

			HCC								
			Yes			No			Total		
PAGE-B		<11	1			274			275		
		≥11	7			197			204		
	Sensitivity			Specificity	PPV		NPV	PLR		NLR	Accuracy
PAGE-B ≥ 11	0.875			0.582	0.034		0.996	2.09		0.214	0.586

PPV: positive predictive value; NPV: negative predictive value; PLR: positive likelihood ratio; NLR: negative likelihood ratio.

**Table 9 viruses-14-00732-t009:** Distribution of patients with and without hepatocellular carcinoma, according to REACH-B.

		HCC	
		Yes	No
REACH-B	≥7.5	13	255
	<7.5	03	411
	≥9.5	10	129
	<9.5	06	537

**Table 10 viruses-14-00732-t010:** Accuracy in HCC prediction, using the cut-off point ≥7.5 and ≥9.5 in the REACH-B score.

	Sensitivity	Specificity	PPV	NPV	PLR	NLR	Accuracy
REACH-B ≥ 7.5	0.812	0.617	0.048	0.992	2.122	0.303	0.621
REACH-B ≥ 9.5	0.625	0.806	0.071	0.988	3.226	0.465	0.802

PPV: positive predictive value; NPV: negative predictive value; PLR: positive likelihood ratio; NLR: negative likelihood ratio.

## Data Availability

All available data is included in the manuscript.

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
