# Peer review of "PAGE-B and REACH-B Predicts the Risk of Developing Hepatocellular Carcinoma in Chronic Hepatitis B Patients from Northeast, Brazil"

_viruses, 2022, doi:10.3390/v14040732_

Round 1

Reviewer 1 Report

he manuscript entitled “PAGE-B and REACH-B, predicts the risk of developing hepatocellular carcinoma in chronic hepatitis B patients from Northeast, Brazil” by Alessandra Porto de Macedo Costa et describes a novel strategy for screening the potential risk of HCC development in HBV infected patients. PAGE-B (platelets, age, sex, and HBV) and REACH-B (risk estimate for hepatocellular carcinoma in chronic hepatitis B]) appears to be an interesting model to predict the risk of developing HCC in HBV patients. Authors have validated the sensitivity and specificity of the PAGE-B and REACH-B in patient population and risk stratification scores based on these models have been shown to be a cost effective strategy. These studies were very systematically conducted, and I recommend that this manuscript may be published without any reservation he manuscript entitled “PAGE-B and REACH-B, predicts the risk of developing hepatocellular carcinoma in chronic hepatitis B patients from Northeast, Brazil” by Alessandra Porto de Macedo Costa et describes a novel strategy for screening the potential risk of HCC development in HBV infected patients. PAGE-B (platelets, age, sex, and HBV) and REACH-B (risk estimate for hepatocellular carcinoma in chronic hepatitis B]) appears to be an interesting model to predict the risk of developing HCC in HBV patients. Authors have validated the sensitivity and specificity of the PAGE-B and REACH-B in patient population and risk stratification scores based on these models have been shown to be a cost effective strategy. These studies were very systematically conducted, and I recommend that this manuscript may be published without any reservation he manuscript entitled “PAGE-B and REACH-B, predicts the risk of developing hepatocellular carcinoma in chronic hepatitis B patients from Northeast, Brazil” by Alessandra Porto de Macedo Costa et describes a novel strategy for screening the potential risk of HCC development in HBV infected patients. PAGE-B (platelets, age, sex, and HBV) and REACH-B (risk estimate for hepatocellular carcinoma in chronic hepatitis B]) appears to be an interesting model to predict the risk of developing HCC in HBV patients. Authors have validated the sensitivity and specificity of the PAGE-B and REACH-B in patient population and risk stratification scores based on these models have been shown to be a cost effective strategy. These studies were very systematically conducted, and I recommend that this manuscript may be published without any reservation he manuscript entitled “PAGE-B and REACH-B, predicts the risk of developing hepatocellular carcinoma in chronic hepatitis B patients from Northeast, Brazil” by Alessandra Porto de Macedo Costa et describes a novel strategy for screening the potential risk of HCC development in HBV infected patients. PAGE-B (platelets, age, sex, and HBV) and REACH-B (risk estimate for hepatocellular carcinoma in chronic hepatitis B]) appears to be an interesting model to predict the risk of developing HCC in HBV patients. Authors have validated the sensitivity and specificity of the PAGE-B and REACH-B in patient population and risk stratification scores based on these models have been shown to be a cost effective strategy. These studies were very systematically conducted, and I recommend that this manuscript may be published without any reservation he manuscript entitled “PAGE-B and REACH-B, predicts the risk of developing hepatocellular carcinoma in chronic hepatitis B patients from Northeast, Brazil” by Alessandra Porto de Macedo Costa et describes a novel strategy for screening the potential risk of HCC development in HBV infected patients. PAGE-B (platelets, age, sex, and HBV) and REACH-B (risk estimate for hepatocellular carcinoma in chronic hepatitis B]) appears to be an interesting model to predict the risk of developing HCC in HBV patients. Authors have validated the sensitivity and specificity of the PAGE-B and REACH-B in patient population and risk stratification scores based on these models have been shown to be a cost effective strategy. These studies were very systematically conducted, and I recommend that this manuscript may be published without any reservation he manuscript entitled “PAGE-B and REACH-B, predicts the risk of developing hepatocellular carcinoma in chronic hepatitis B patients from Northeast, Brazil” by Alessandra Porto de Macedo Costa et describes a novel strategy for screening the potential risk of HCC development in HBV infected patients. PAGE-B (platelets, age, sex, and HBV) and REACH-B (risk estimate for hepatocellular carcinoma in chronic hepatitis B]) appears to be an interesting model to predict the risk of developing HCC in HBV patients. Authors have validated the sensitivity and specificity of the PAGE-B and REACH-B in patient population and risk stratification scores based on these models have been shown to be a cost effective strategy. These studies were very systematically conducted, and I recommend that this manuscript may be published without any reservation he manuscript entitled “PAGE-B and REACH-B, predicts the risk of developing hepatocellular carcinoma in chronic hepatitis B patients from Northeast, Brazil” by Alessandra Porto de Macedo Costa et describes a novel strategy for screening the potential risk of HCC development in HBV infected patients. PAGE-B (platelets, age, sex, and HBV) and REACH-B (risk estimate for hepatocellular carcinoma in chronic hepatitis B]) appears to be an interesting model to predict the risk of developing HCC in HBV patients. Authors have validated the sensitivity and specificity of the PAGE-B and REACH-B in patient population and risk stratification scores based on these models have been shown to be a cost effective strategy. These studies were very systematically conducted, and I recommend that this manuscript may be published without any reservation 

Author Response

Dear Reviewer 1

We would like to thank you for your comments about this manuscript. 

Sincerely

Reviewer 2 Report

Comments:

Up to now, HBV associated HCC is a serious global public health issue, especially in the HBV high prevalence area. It is really need an accuracy of predicting the risk of developing HCC in patients with CHB to screen this high risk population.

The authors want to identify the accuracy of the PAGE-B and REACH-B scores in predicting the risk of developing HCC in patients with chronic hepatitis B regularly followed up at a reference service in the State of Maranhão. The studies with a historical, longitudinal, retrospective cohort identified that PAGE-B and REACH-B scores were able to predict the risk of developing HCC in the studied population. Their studies identified that the PAGE-B and REACH-B scores are useful to reduce costs associated with HCC screening.

This is a very interested story for readers and I suggest the Journal accept this manuscript to publish.

Author Response

Dear Reviewer 2,

We would like to thank you for your comments about the manuscript.

Sincerely,

Reviewer 3 Report

In the manuscript, Costa APM et al used a ROC curve to evaluate the accuracy and applicability of PAGE-B and REACH-B score in predicating the incidence rate of hepatocellular carcinoma in chronic hepatitis B patients. The authors demonstrated that the prediction model of PAGE-B and REACH-B score have increased ROC curve data adherence with the increased temporal scenario. Analysis with 978 patients’ medical records shown that PAGE-B and REACH-B score have considerable accuracy to predict HCC. This manuscript made a contribution in confirming the application of PAGE-B and REACH-B score in predicating HCC from Northeast, Brazil but also has a number of shortcomings and potential technical problems that need to be addressed.                

  • In table 5, the median 47 and 60.5 missed percentage in the age term.
  • Patients those don’t have a family history of HBV or HCC are more possible to develop HCC is interesting and odd, the readers may have much interest in this and should better explain or discuss it in discussion section.   
  • The resolution of FIG 1 and FIG 2 are not enough for publication.
  • Some of the references are ill-formed, for example the reference 3, 14, 31, 42. Check all the references make sure they are right.
  • The authors should provide the more detail information of survival ROC package in the materials and methods section.
  • The function of REACH-B score is to assesses the risk of developing HCC at 3, 5 and 10 years in untreated patients, while when analyze the accuracy of REACH-B, the authors only consider the patients’ medical records with medical follow-up for at least 3 years, the authors failed to verify the accuracy of PEACH-B in predicating the risk of developing HCC at 5 and 10 years.

Author Response

Dear Reviewer 3

In order to your comments, we have better improved the manuscript.

  1. In table 5, the median 47 and 60.5 missed percentage in the age term.

Answer: We correct this information. This is about median not mean.

2. Patients those don’t have a family history of HBV or HCC are more possible to develop HCC is interesting and odd, the readers may have much interest in this and should better explain or discuss it in discussion section.  

Answer: In this case, In this case, we believe that this result is due to the fact that most patients do not have a family history of HBV. 

Answer: We have included an statement about this in the discussion section.

3. The resolution of FIG 1 and FIG 2 are not enough for publication.

Asnwer: We have improved the quality of these figures

4. Some of the references are ill-formed, for example the reference 3, 14, 31, 42. Check all the references make sure they are right.

Asnwer: We have corrected the references. 

5. The authors should provide the more detail information of survival ROC package in the materials and methods section.

Asnwer: We have better described these information in methods section

6. The function of REACH-B score is to assesses the risk of developing HCC at 3, 5 and 10 years in untreated patients, while when analyze the accuracy of REACH-B, the authors only consider the patients’ medical records with medical follow-up for at least 3 years, the authors failed to verify the accuracy of PEACH-B in predicating the risk of developing HCC at 5 and 10 years.

Asnwer: Please, notice that we have included patients with at least 3 years of follow-up. All patient with at least 3 years were  included.

Sincerely,

Round 2

Reviewer 3 Report

The revised manuscript is ready for publication.